# Dynamic Stiffness Enhancement of the Quadcopter Control System

**Wu-Sung Yao * and Chun-Yi Lin**

Department of Mechatronics Engineering, National Kaohsiung University of Science and Technology, Kaohsiung City 824, Taiwan; i109109110@nkust.edu.tw
* Correspondence: wsyao@nkust.edu.tw; Tel.: +886-7-6011000 (ext. 37616)

**Abstract:** Vibrations often result in fatigue breakage in quadcopter flight operations. Reducing the vibration effect is the main important issue for quadcopter flight. Enhanced dynamic stiffness is required in the quadcopter control system for the vibration rejection ability. In this paper, the dynamic stiffness of the quadcopter control system is constructed and is used as an index for the performance of the quadcopter control system to resist an external oscillatory load. To rapidly reduce the vibration effect, a repetitive controller is introduced in the quadcopter control system, where the direct dynamic stiffness and the quadrature dynamic stiffness with variable frequencies are proposed. A theoretical model of the dynamic stiffness of the quadcopter control system is established by analyzing the definition of dynamic stiffness. Simulated and experimental results show that the magnitudes of the direct dynamic stiffness with the proposed repetitive controller are larger than those without the repetitive controller. The quadrature dynamic stiffness with the proposed repetitive controller is relatively smooth compared to that without the proposed repetitive controller, which can be used to verify the stability being improved by the designed repetitive controller. In addition, the magnitudes of the loss factor of the quadcopter control system with the repetitive controller are lower than those without the repetitive controller.

**Keywords:** vibration rejection; dynamic stiffness; quadcopter; repetitive controller; direct dynamic stiffness; quadrature dynamic stiffness

## 1. Introduction

Quadcopters have become a research topic in recent years because of their special functionality and industrial requirements. Quadcopters can not only increase the freedom of movement but also can be remotely controlled, so industrial applications have increased, such as for surveillance, search and rescue, building inspection, and some other applications. The quadcopter system has 6 degrees of freedom of movement in space (translation and rotation along three coordinate axes) but only 4 control degrees of freedom (the rotation speed of the four motors), which is called an under-actuated control system. It is well known that a complete actuated system is when the control degrees of freedom are equal to the movement degrees of freedom. However, for the attitude control (rotational motion along three axes), it is indeed fully driven. Compared with helicopters, quadcopters can achieve fewer flight attitudes, but basic forward, backward, translation and other states can be operated. However, the mechanical structure of the quadcopter is far simpler than that of the helicopter, and the maintenance and substitution costs are also very small, which gives the quadcopter a greater application advantage than the helicopter [1,2].

In order to achieve a stable flight operator, the quadcopter is supplied with three-directional gyroscopes and three-axis acceleration sensors to form an inertial navigation module, which can reckon the quadcopter's attitude, acceleration, and angular velocity relative to the ground. The flight controller can be used to obtain the rotational force and lift required to maintain the motion state through an algorithm and uses the electronic controller to ensure that the motor can output the appropriate force.

With two pairs of motors in opposite directions, the counter-torque to the fuselage can be balanced. For the output power of the four motors to be increased, the rotor speed and the total pulling force can be increased. The total pulling force should be enough to supply the whole quadcopter's weight. The rotorcraft will rise vertically from the ground; on the contrary, the output power of the four motors will be reduced. In addition, the quadrotor will fall vertically until it lands in a balanced manner, realizing vertical movement along the *z*-axis. Without the external disturbance, the lift generated by the rotor is equal to the quadcopter's weight, and the quadcopter will remain in a hovering state. The key to vertical motion is to ensure that the rotation speed of the four rotors increases or decreases synchronously.

At present, the most common algorithm in the application of quadcopters is to use complementary filtering: that is, to calculate the angle change by combining the output of the acceleration sensor and the gyroscope. In addition, the environment of the quadcopter operation determines the MEMS sensor, which must be suitable for various harsh conditions while obtaining high-precision output. For example, the ideal output of a gyroscope is to only respond to changes in angular velocity, but in fact, due to the limitations of design and craftsmanship, gyroscopes are also sensitive to acceleration, which is the deg/sec/g indicator on gyroscope datasheets. For the application of a multi-rotor aircraft, this indicator is particularly important, because the motor in the aircraft generally brings relatively strong vibration. The change in the output of the gyroscope will cause the change in the angle, and the motor will malfunction. With the presence of external factors, vibration can be found in a quadcopters' flight to cause a non-uniform flow field. In practice, the actual quadcopter lift force is made up of harmonics related to the angular velocity of the quadcopter propeller.

Many related works in the literature can be found to handle the problems of the disturbance, uncertainties, and motor faults for the quadcopter control system. In [3], the issue of the quadcopter with vibration mitigation is proposed. In [4], an adaptive sliding mode control is proposed to solve the problem of uncertainty and external disturbances for quadcopter flight. In [5], the detection and diagnosis of motor and propeller degradation is raised to reduce the vibration of the quadcopter operation. In [6], to solve the problem of motor faults, the fault-tolerant control is addressed. In [7], a model predictive controller and the disturbance observer are given to stable the quadcopter. In [8], a robust hybrid nonlinear control is applied to the quadcopter with the cross-coupling disturbance. In [9], to reduce the effect of load uncertainties, an observer-based attitude stabilization mechanism is proposed. In [10], an aerodynamic model is proposed to simulate the model of the external disturbance of the quadcopter. In [11], with parametric uncertainties and external disturbances, an adaptive fast finite-time control is adapted for a tilting quadcopter. In [12], a cooperative path following control is given to improve the stability of the quadcopter with unknown external disturbances. In [13], a bidirectional fuzzy brain emotional learning controller is proposed to solve the problem of the payload uncertainties and disturbances. In [14], a robust control with a disturbance observer is considered to improve the stability of the quadcopter. In [15], a finite-time control method for a multirotor UAV is proposed with parameter uncertainties and external disturbances. In [16], a fault-tolerant control is given to the damaged propellers. In [17], with the vibration of the multi-rotor arms, a fault detection using artificial intelligence is proposed. In [18], wind measurement and simulation methods for multi-rotor UAVs are proposed to achieve a stable operation. In [19], with motor fault and external disturbance, an adaptive attitude control for multi-rotor UAVs is given. In [20], a disturbance observer-based controller is proposed with parametric uncertainties and wind disturbance. Therefore, according to related works in the literature, how to reduce the vibration generated by external disturbance, wind disturbance, and motor/propeller faults is the key point for the quadcopter operation.

In general, the ability of lower frequency disturbance rejection can be regarded as static stiffness, which can be achieved by a high mathematical design [21]. Therefore, to achieve better performance of the quadcopter control system, the dynamic stiffness [21–23] should be enhanced. In this paper, a methodology to analyze the dynamic stiffness of the

quadcopter control system is proposed. The dynamic stiffness is used to be an index for the performance of the quadcopter control system to resist an external oscillatory load. The proposed dynamic stiffness is varying under excitations at different frequencies. The outline of this paper is developed as follows. Section 2 contains the mathematical modeling of the quadcopter. In Section 3, the dynamic stiffness of the quadcopter control system with a repetitive controller is proposed. Simulations and experimental results are performed in Section 4. Finally, concluding remarks are stated in Section 5.

## 2. System Description and Modeling

The quadcopter's modeling is shown in Figure 1, where the four motors are used to force the machine. The inertial frame $\xi = (x, y, z)$ with three axes of x, y, and z is defined, while the angular position can be obtained as $\eta = (\phi, \theta, \varphi)$. The three Euler angles of $\phi$ (defined as roll angle), $\theta$ (defined as pitch angle), and $\psi$ (defined as yaw angle) are represented as the rotations around the x, y, and z-axis, respectively. A vector $\mathbf{q} = (\xi, \eta)$ is obtained by the linear vector of $\xi = (x, y, z)$ and the angular position vector of $\eta = (\phi, \theta, \varphi)$. In the frame, the linear velocities of $\mathbf{V} = (v_x, v_y, v_z)$ and the angular velocities of $\mathbf{v} = (p, q, r)$ are defined. The rotation matrix of $\mathbf{R}$ from the body frame to the inertial frame can be determined by

$$\mathbf{R} = \begin{bmatrix} c_\varphi c_\theta & c_\varphi s_\theta s_\phi - s_\varphi c_\phi & c_\varphi s_\theta c_\phi + s_\varphi s_\phi \\ s_\varphi c_\theta & s_\varphi s_\theta s_\phi + c_\varphi c_\phi & s_\varphi s_\theta c_\phi - c_\varphi s_\phi \\ -s_\theta & c_\theta s_\phi & c_\theta c_\phi \end{bmatrix} \tag{1}$$

where $s_i = \sin i$ and $c_i = \cos i$ for $i = \phi, \theta, \varphi$ are defined.

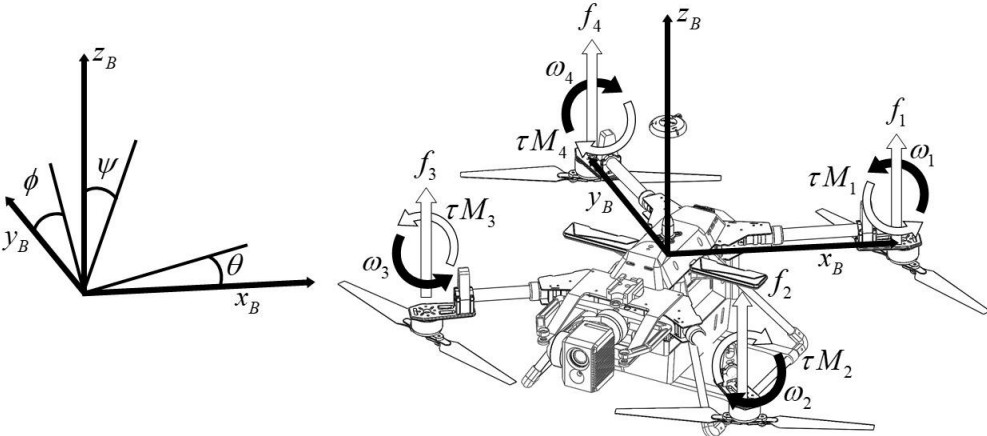

**Figure 1.** Inertial and body frames of a quadcopter.

$\mathbf{W}_\eta$ is defined as the transformation matrix for angular velocities from the inertial frame to the body frame. Then, we have

$$\mathbf{W}_\eta = \begin{bmatrix} 1 & 0 & -s_\theta \\ 0 & c_\phi & c_\theta s_\phi \\ 0 & -s_\phi & c_\theta c_\phi \end{bmatrix} \tag{2}$$

and

$$\mathbf{v} = \begin{bmatrix} p \\ q \\ r \end{bmatrix} = \mathbf{W}_\eta \begin{bmatrix} \frac{d\phi}{dt} \\ \frac{d\theta}{dt} \\ \frac{d\varphi}{dt} \end{bmatrix} \tag{3}$$

Assume that the four arms of the quadcopter are aligned to the *x*- and *y*-axes of the device body. Thus, the inertia matrix of $\mathbf{I} = diag(I_{xx}, I_{yy}, I_{zz})$ is a diagonal matrix with $I_{xx} = I_{yy}$. The angular velocity of the rotor axis is given as $\omega_i$, and $f_i$ is the generated force

in the direction of the rotor axis. $\tau_{M_i}$ is the torque of the rotor axis, where $i = 1, 2, 3, 4$ are for the four-rotor axis. Then, we have $f_i = k\omega_i^2$ and $\tau_{M_i} = b\omega_i^2 + I_M \frac{d\omega_i}{dt}$, where $k$ is the lift constant, $b$ is the drag constant, and the rotor inertia moment is defined as $I_M$. In the $z$-axis of the body frame, the combined thrust of rotors is given as $T$. The torque of $\boldsymbol{\tau}_B = \begin{pmatrix} \tau_\phi & \tau_\theta & \tau_\varphi \end{pmatrix}$ is defined in the direction of $(\phi, \theta, \varphi)$. Therefore, we have

$$T = \sum_{i=1}^{4} f_i = k \sum_{i=1}^{4} \omega_i^2 \tag{4}$$

and

$$\boldsymbol{\tau}_B = \begin{bmatrix} \tau_\phi \\ \tau_\theta \\ \tau_\varphi \end{bmatrix} = \begin{bmatrix} lk\left(-\omega_2{}^2 + \omega_4{}^2\right) \\ lk\left(-\omega_1{}^2 + \omega_3{}^2\right) \\ \sum\limits_{i=1}^{4} \tau_{M_i} \end{bmatrix} \tag{5}$$

where $l$ is the distance between the mass center and the rotor.

Therefore, the dynamics characteristic is determined by Newton–Euler equations. In the body frame, $m \frac{d\mathbf{V}_B}{dt}$ is defined as the force for the mass acceleration, $\mathbf{v} \times m\mathbf{V}_B$ is defined by the centrifugal force being equal to the gravity of $\mathbf{R}^T\mathbf{G}$. Then, the whole thrust $\mathbf{T}_B$ generated by the rotors motor can be obtained by

$$m \frac{d\mathbf{V}_B}{dt} + \mathbf{v} \times m\mathbf{V}_B = \mathbf{R}^T\mathbf{G} + \mathbf{T}_B \tag{6}$$

Assume that the centrifugal force is neglected. Thus, (7) can be simplified as

$$m \frac{d^2\boldsymbol{\xi}}{dt^2} = \mathbf{G} + \mathbf{R}\mathbf{T}_B \tag{7}$$

In the body frame, the angular acceleration of the inertia can be obtained by $\mathbf{I}\frac{d\mathbf{v}}{dt}$, the centripetal forces and the gyroscopic forces are determined by $\mathbf{v} \times (\mathbf{I}\mathbf{v})$ and $\boldsymbol{\Gamma}$, respectively, and $\boldsymbol{\tau}$ is represented by the external torque. Therefore, we have

$$\mathbf{I}\frac{d\mathbf{v}}{dt} + \mathbf{v} \times (\mathbf{I}\mathbf{v}) + \boldsymbol{\Gamma} = \boldsymbol{\tau} \tag{8}$$

The angular accelerations in the inertial frame can be given as

$$\frac{d^2\boldsymbol{\eta}}{dt^2} = \frac{d\left(\mathbf{W}_\eta{}^{-1}\mathbf{v}\right)}{dt} = \frac{d\mathbf{W}_\eta{}^{-1}}{dt}\mathbf{v} + \mathbf{W}_\eta{}^{-1}\frac{d\mathbf{v}}{dt} \tag{9}$$

The Lagrangian $\Lambda$ can be given as

$$\Lambda\left(q, \frac{dq}{dt}\right) = \frac{m}{2}\left(\frac{d\boldsymbol{\xi}}{dt}\right)^T \frac{d\boldsymbol{\xi}}{dt} + \frac{\mathbf{v}^T\mathbf{I}\mathbf{v}}{2} - mgz \tag{10}$$

The external forces and torques of the device can be determined by Euler–Lagrange equations, i.e.,

$$\begin{bmatrix} \mathbf{f} \\ \boldsymbol{\tau} \end{bmatrix} = \frac{d}{dt}\left(\frac{\partial\Lambda}{\partial\left(\frac{dq}{dt}\right)}\right) - \frac{\partial\Lambda}{\partial q} \tag{11}$$

The Jacobian matrix of $\mathbf{J}(\boldsymbol{\eta})$ from $\mathbf{v}$ to $\frac{d\boldsymbol{\eta}}{dt}$ is

$$\mathbf{J}(\boldsymbol{\eta}) = \mathbf{W}_\eta{}^T\mathbf{I}\mathbf{W}_\eta = \begin{bmatrix} I_{xx} & 0 & -I_{xx}\mathsf{s}_\theta \\ 0 & I_{yy}\mathsf{c}_\varphi^2 + I_{zz}\mathsf{s}_\phi^2 & \left(I_{yy} - I_{zz}\right)\mathsf{c}_\phi\mathsf{s}_\phi\mathsf{c}_\theta \\ -I_{xx}\mathsf{s}_\theta & \left(I_{yy} - I_{zz}\right)\mathsf{c}_\phi\mathsf{s}_\phi\mathsf{c}_\theta & I_{xx}\mathsf{s}_\theta^2 + I_{yy}\mathsf{s}_\phi^2\mathsf{c}_\theta^2 + I_{zz}\mathsf{c}_\phi^2\mathsf{c}_\theta^2 \end{bmatrix} \tag{12}$$

Thus, in the inertial frame, the rotational energy $E_{rot}$ is obtained by

$$E_{rot} = \frac{1}{2}\mathbf{v}^T \mathbf{I}\mathbf{v} = \frac{1}{2}\left(\frac{d^2\boldsymbol{\eta}}{dt^2}\right)^T \mathbf{J}\left(\frac{d^2\boldsymbol{\eta}}{dt^2}\right) \tag{13}$$

The external angular force is the torque outputs of the rotor motors, i.e.,

$$\boldsymbol{\tau} = \boldsymbol{\tau}_B = \mathbf{J}\frac{d^2\boldsymbol{\eta}}{dt^2} + \frac{d\mathbf{J}}{dt}\frac{d\boldsymbol{\eta}}{dt} - \frac{1}{2}\frac{\partial}{\partial\boldsymbol{\eta}}\left[\left(\frac{d\boldsymbol{\eta}}{dt}\right)^T \mathbf{J}\left(\frac{d\boldsymbol{\eta}}{dt}\right)\right] = \mathbf{J}\frac{d^2\boldsymbol{\eta}}{dt^2} + C\left(\boldsymbol{\eta}, \frac{d\boldsymbol{\eta}}{dt}\right)\frac{d\boldsymbol{\eta}}{dt} \tag{14}$$

where the matrix $C\left(\boldsymbol{\eta}, \frac{d\boldsymbol{\eta}}{dt}\right)$ being the Coriolis term contains the gyroscopic and centripetal terms. Then, we have

$$C\left(\boldsymbol{\eta}, \frac{d\boldsymbol{\eta}}{dt}\right) = \begin{bmatrix} 0 & \begin{bmatrix} (I_{yy} - I_{zz})\left(\frac{d\theta}{dt}c_\phi s_\phi + \frac{d\varphi}{dt}s_\phi^2 c_\theta\right) \\ +(I_{zz} - I_{yy})\left(\frac{d\varphi}{dt}c_\phi^2 c_\theta - \frac{d\varphi}{dt}I_{xx}c_\theta\right) \end{bmatrix} & (I_{zz} - I_{yy})\frac{d\varphi}{dt}c_\phi s_\phi c_\theta^2 \\ \begin{bmatrix} (I_{zz} - I_{yy})\left(\frac{d\theta}{dt}c_\phi s_\phi + \frac{d\varphi}{dt}s_\phi c_\theta\right) \\ +(I_{yy} - I_{zz})\left(\frac{d\varphi}{dt}c_\phi^2 c_\theta + \frac{d\varphi}{dt}I_{xx}c_\theta\right) \end{bmatrix} & (I_{zz} - I_{yy})\frac{d\varphi}{dt}c_\phi s_\phi & \begin{bmatrix} -I_{xx}\frac{d\varphi}{dt}s_\theta c_\theta + I_{yy}\frac{d\varphi}{dt}s_\phi^2 s_\theta c_\theta \\ +I_{zz}\frac{d\varphi}{dt}c_\phi^2 s_\theta c_\theta \end{bmatrix} \\ (I_{yy} - I_{zz})\frac{d\varphi}{dt}c_\theta^2 s_\phi c_\varphi - I_{xx}\frac{d\theta}{dt}c_\theta & \begin{bmatrix} (I_{zz} - I_{yy})\left(\frac{d\theta}{dt}c_\phi s_\phi s_\theta + \frac{d\phi}{dt}s_\phi^2 c_\theta\right) \\ +(I_{yy} - I_{zz})\left(\frac{d\phi}{dt}c_\phi^2 c_\theta\right) + I_{xx}\frac{d\varphi}{dt}s_\theta c_\theta \\ -I_{yy}\frac{d\varphi}{dt}s_\phi^2 s_\theta c_\theta - I_{zz}\frac{d\varphi}{dt}c_\phi^2 s_\theta c_\theta \end{bmatrix} & \begin{bmatrix} (I_{yy} - I_{zz})\frac{d\phi}{dt}c_\phi s_\phi c_\theta^2 + I_{xx}\frac{d\theta}{dt}s_\theta c_\theta \\ -I_{yy}\frac{d\theta}{dt}s_\phi^2 s_\theta c_\theta - I_{zz}\frac{d\theta}{dt}c_\phi^2 s_\theta c_\theta \end{bmatrix} \end{bmatrix} \tag{15}$$

From (17), we have

$$\frac{d^2\boldsymbol{\eta}}{dt^2} = \mathbf{J}^{-1}\left(\boldsymbol{\tau}_B - C\left(\boldsymbol{\eta}, \frac{d\boldsymbol{\eta}}{dt}\right)\frac{d\boldsymbol{\eta}}{dt}\right) \tag{16}$$

The simplified model of the quadcopter can be obtained as

$$m\frac{d^2\mathbf{X}}{dt^2} + \mathbf{D}\frac{d\mathbf{X}}{dt} + m\mathbf{G} = \mathbf{T} \tag{17}$$

where $\mathbf{X} = \begin{bmatrix} x & y & z \end{bmatrix}^T$, $\mathbf{D} = diag(D_x, D_y, D_z)$, $\mathbf{G} = \begin{bmatrix} 0 & 0 & g \end{bmatrix}^T$, and $\mathbf{T} = \begin{bmatrix} (c_\varphi s_\theta c_\phi + s_\varphi s_\phi)T \\ (s_\varphi s_\theta c_\phi - c_\varphi s_\phi)T \\ (c_\theta c_\phi)T \end{bmatrix}$ are given. $D_x$, $D_y$ and $D_z$ are the drag force coefficients for velocities in $(x, y, z)$.

## 3. Dynamic Stiffness Analysis of the Quadcopter Control System

A closed loop control system is shown in Figure 2, where $G(s)$ and $K(s)$ are the controlled plant and loop controller, respectively. In general, $d$ is the system vibration disturbance, $y$ is the system output, $u$ is the controller output, and $e$ is the error. Note that $P(s) = K(s)G(s)$. Let the frequency responses of $P(j\omega) = P_1(\omega) + jP_2(\omega)$ and $K(j\omega) = C_1(\omega) + jC_2(\omega)$ be given, where we have

$$\begin{bmatrix} P_1(\omega) = G_1C_1(\omega) - G_2C_2(\omega) \\ P_2(\omega) = G_1C_2(\omega) + G_1C_2(\omega) \end{bmatrix} \tag{18}$$

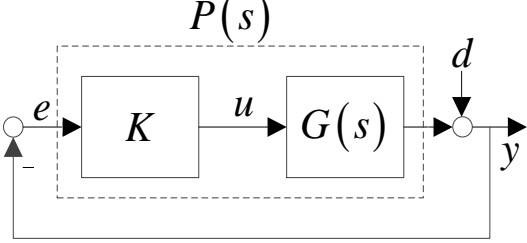

**Figure 2.** Closed-loop control system.

Therefore, the dynamic stiffness of Figure 2 can be denoted by $D(s) = 1 + P(s)$. We have $D(j\omega) = D_1(\omega) + jD_2(\omega)$, and $[D(j\omega)]^{-1}$ is defined as the dynamic compliance. Note that $D_1(\omega) = 1 + P_1(\omega)$ and $D_2(\omega) = P_2(\omega)$ are defined as direct dynamic stiffness and the quadrature dynamic stiffness, respectively. Note that $D_1(\omega)$ can be represented as the conservative properties of the system, and $D_2(\omega)$ can be represented as the dissipative properties. Then, the magnitude of the dynamic stiffness $D(j\omega)$ can be given as $|D(j\omega)| = \sqrt{[1 + P_1(\omega)]^2 + [P_2(\omega)]^2}$, which can be rewritten as $D(j\omega) = D_1(\omega)[1 + jH(\omega)]$, and $H(\omega)$ can be defined as the loss factor of the control system, i.e.,

$$H(\omega) = \frac{D_2(\omega)}{D_1(\omega)} = \frac{P_2(\omega)}{1 + P_1(\omega)} \tag{19}$$

For the required performance, it can be expected that $D_1(\omega)$ is increasing with $\omega$ and $H(\omega)$ is decreasing with $\omega$. From the above analysis, the dynamic stiffness of the closed-loop control system in Figure 3 is indeed more than that without a loop controller.

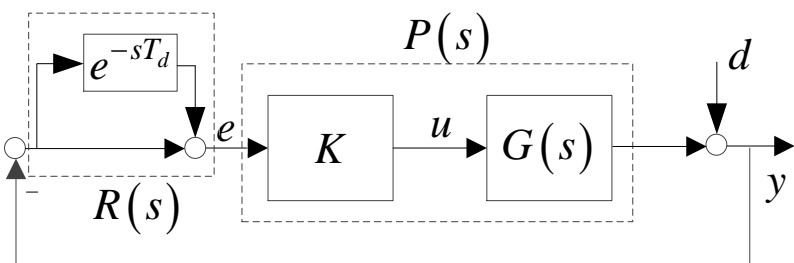

**Figure 3.** Repetitive control system.

For the vibration rejection, a repetitive control system is induced, as shown in Figure 3. Assume that the vibration disturbance $d$ is a single period of $T_d$. The dynamic stiffness of Figure 3 can be denoted by

$$D_r(s) = 1 + R(s)P(s) = 1 + \frac{P(s)}{1 - e^{-sT_d}}. \tag{20}$$

Then, the frequency response of $D_r(j\omega)$ can be given as $D_r(j\omega) = D_{r1}(\omega) + jD_{r2}(\omega)$. Note that $D_{r1}(\omega) = 1 + P_1(\omega)\sum\limits_{k=0}^{n} c_k + P_2(\omega)\sum\limits_{k=0}^{n} s_k$ and $D_{r2}(\omega) = P_2(\omega)\sum\limits_{k=0}^{n} c_k - P_1(\omega)\sum\limits_{k=0}^{n} s_k$ are represented as the direct dynamic stiffness and the quadrature dynamic stiffness, respectively, where $c_k = \cos(k\omega T_d)$ and $s_k = \sin(k\omega T_d)$ are defined.

The loss factor $H_r(\omega)$ of Figure 3 can be given as

$$H_r = \frac{D_{r2}(\omega)}{D_{r1}(\omega)} = \frac{P_2(\omega)\sum\limits_{k=0}^{n} c_k - P_1(\omega)\sum\limits_{k=0}^{n} s_k}{1 + P_1(\omega)\sum\limits_{k=0}^{n} c_k + P_2(\omega)\sum\limits_{k=0}^{n} s_k} \tag{21}$$

As $\omega = \frac{2k\pi}{T_d} = k\omega_d$, $k \in N$, the loss factor $H_r(k\omega_d)$ can be given as

$$H_r(k\omega_d) = \frac{(1+n)P_2(k\omega_d)}{1 + (1+n)P_1(k\omega_d)} \tag{22}$$

Due to the poles of $R(s) = \frac{1}{1 - e^{-sT_d}}$ being located at an imaginary axis, to enhance the stability of the repetitive control system in Figure 3, a low-pass filter $Q(s)$ is induced as shown in Figure 4, where $\|Q(j\omega)\| \approx 1$ for $\omega \leq \omega_q$ is given.

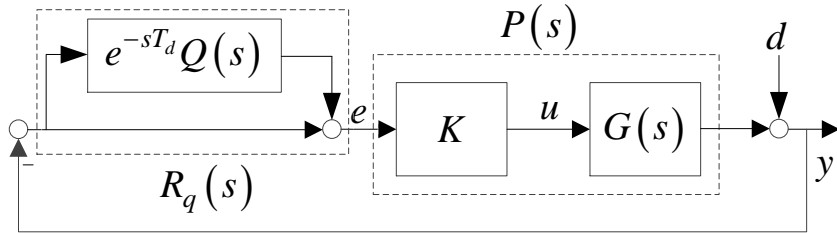

**Figure 4.** Repetitive control system with $Q(s)$.

The dynamic stiffness of Figure 4 can be denoted by

$$D_{qr}(s) = 1 + R_q(s)P(s) = 1 + \frac{P(s)}{1 - e^{-sT_d}Q(s)} \tag{23}$$

Its frequency response can be given as

$$D_{qr}(j\omega) = 1 + P(j\omega)\left[1 + {}_qe^{-j\omega T_d} + {}_qe^{-2j\omega T_d} + \ldots\right] \tag{24}$$

where ${}_qe^{-jn\omega T_d} = e^{-jn\omega T_d}[Q(j\omega)]^n$, $n = 1, 2, 3, \ldots$ is defined.
Note that $Q(j\omega) = Q_1(\omega) + jQ_2(\omega)$ is given.
In addition, we have

$$\begin{bmatrix} D_{qr}(j\omega) = 1 + (P_1(\omega) + jP_2(\omega))\left[1 + \sum_{k=1}^{n}\left[e^{-j\omega T_d}Q(j\omega)\right]^k\right] \\ = 1 + (P_1(\omega) + jP_2(\omega))\left[\sum_{k=0}^{n}\left[(\cos(\omega T_d) - j\sin(\omega T_d))(Q_1(\omega) + jQ_2(\omega))\right]^k\right] \\ = 1 + \sum_{k=0}^{n}(P_1(\omega) + jP_2(\omega))(\widetilde{c} - j\widetilde{s})^k \end{bmatrix} \tag{25}$$

where $\widetilde{c} = c_1 Q_1(\omega) + s_1 Q_2(\omega)$ and $\widetilde{s} = s_1 Q_1(\omega) - c_1 Q_2(\omega)$ are given.

We also have $|Q(j\omega)| = \sqrt{\widetilde{c}^2 + \widetilde{s}^2}$. Note that $D_{qr1}(\omega) = 1 + P_1(\omega)\sum_{k=0}^{n}\widetilde{c}_k + P_2(\omega)\sum_{k=0}^{n}\widetilde{s}_k$

and $D_{qr2}(\omega) = P_2(\omega)\sum_{k=0}^{n}\widetilde{c}_k - P_1(\omega)\sum_{k=0}^{n}\widetilde{s}_k$ are defined as the direct dynamic stiffness and the quadrature dynamic stiffness, respectively.

The loss factor $H_{qr}(\omega)$ of Figure 4 can be given as

$$H_{qr} = \frac{D_{qr2}(\omega)}{D_{qr1}(\omega)} = \frac{P_2(\omega)\sum_{k=0}^{n}\widetilde{c}_k - P_1(\omega)\sum_{k=0}^{n}\widetilde{s}_k}{1 + P_1(\omega)\sum_{k=0}^{n}\widetilde{c}_k + P_2(\omega)\sum_{k=0}^{n}\widetilde{s}_k} \tag{26}$$

As $\omega = \frac{2k\pi}{T_r} = k\omega_r$, $k \in N$, the loss factor $H_{qr}(k\omega_r)$ can be given as

$$H_{qr}(k\omega_r) = \frac{P_2(k\omega_r)\sum_{k=0}^{n}[Q_1(k\omega_r)]^k - P_1(k\omega_r)\sum_{k=0}^{n}[-Q_2(k\omega_r)]^k}{1 + P_1(k\omega_r)\sum_{k=0}^{n}[Q_1(k\omega_r)]^k + P_2(\omega)\sum_{k=0}^{n}[-Q_2(k\omega_r)]^k} \tag{27}$$

A general multi-periodic disturbance of $d(t) = \sum_{i=1}^{m}d_i(t)$ is given, where $d_i(t)$ is a periodic signal with period $T_{di}$, $\forall i = 1, 2, \ldots, m$. Therefore, we have a repetitive control system with $d(t) = \sum_{i=1}^{m}d_i(t)$, as shown in Figure 5.

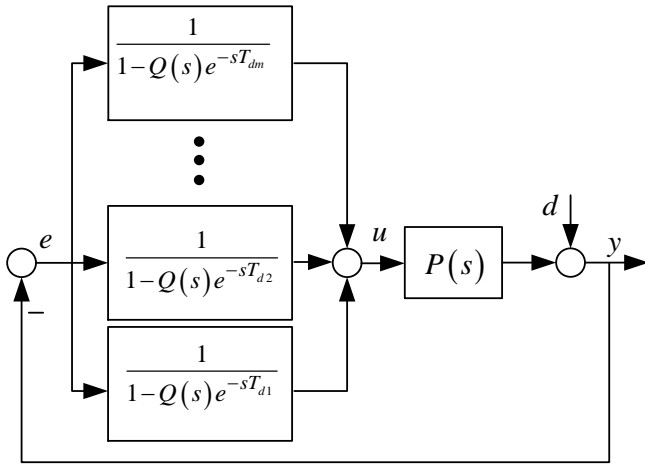

**Figure 5.** Repetitive control system with $d(t) = \sum\limits_{i=1}^{m} d_i(t)$.

The repetitive controller is given as $R_{qm}(s) = \sum\limits_{i=1}^{m} \frac{1}{1-Qe^{-sT_{di}}}$, and its dynamic stiffness can be given as $\hat{D}_{qr}(s) = 1 + P(s) \sum\limits_{i=1}^{m} \frac{1}{1-Q(s)e^{-sT_{di}}}$, which can be rewritten in the frequency domain, i.e.,

$$\hat{D}_{qr}(j\omega) = 1 + \sum\limits_{k=0}^{n} (P_1(\omega) + jP_2(\omega))(\hat{c} - j\hat{s})^k \tag{28}$$

where $\hat{c} = Q_1(\omega) \sum\limits_{i=1}^{m} \cos(\omega T_{di}) + Q_2(\omega) \sum\limits_{i=1}^{m} \sin(\omega T_{di})$ and $\hat{s} = Q_1(\omega) \sum\limits_{i=1}^{m} \sin(\omega T_{di}) - Q_2(\omega) \sum\limits_{i=1}^{m} \cos(\omega T_{di})$ are given.

Note that $\hat{D}_{qr1}(\omega) = 1 + P_1(\omega) \sum\limits_{k=0}^{n} \hat{c}^k + P_2(\omega) \sum\limits_{k=0}^{n} \hat{s}^k$ and $\hat{D}_{qr2}(\omega) = P_2(\omega) \sum\limits_{k=0}^{m} \hat{c}^k - P_1(\omega) \sum\limits_{k=0}^{m} \hat{s}^k$ are defined as the direct dynamic stiffness and the quadrature dynamic stiffness, respectively.

The loss factor $\hat{H}_{qr}(\omega)$ of Figure 5 can be given as

$$\hat{H}_{qr} = \frac{\hat{D}_{qr2}(\omega)}{\hat{D}_{qr1}(\omega)} = \frac{P_2(\omega) \sum\limits_{k=0}^{n} \hat{c}^k - P_1(\omega) \sum\limits_{k=0}^{n} \hat{s}^k}{1 + P_1(\omega) \sum\limits_{k=0}^{n} \hat{c}^k + P_2(\omega) \sum\limits_{k=0}^{n} \hat{s}^k} \tag{29}$$

## 4. Simulated Results Analysis

To analyze the dynamic stiffness of the quadcopter control system, an illustrated example of a quadcopter is shown in Figure 6, where its specifications are listed in Table 1. The controlled plant for the given quadcopter is measured by an analyzer, as shown in Figure 7, where the control plant of $G(s) = \frac{3.329s^2 + 340.19s + 42051}{s^3 + 71.05s^2 + 5248.8s + 56.31}$ can be calculated by the curve fitting. A gain of $K = 0.95$ is pre-determined for the closed loop of $P(s)/(1+P(s))$, which is stable, where $P(s) = KG(s)$ is given. In the designed case, the low-pass filter of $Q(s) = \frac{1}{\frac{s^2}{\omega_q^2} + \frac{2\zeta_q s}{\omega_q} + 1}$, $\omega_q = 20$ Hz, and $\zeta_q = \sqrt{2}/2$ can be designed to achieve the performance of the vibration rejection [24]. Under the proposed quadcopter flight operation, the vibration is measured as shown in Figure 8, where the harmonics of the measured results can be given as 3–11 Hz.

In this study, the flight controller mainly assists the controlled body, while the gimbal image stabilization controller provides the source of vibration interference during flight. The vibration of the gimbal provides a source of interference for the controlled object of the drone to perform experiments. As shown in Figure 6, the red arrow is the gimbal image stabilization controller, which simulates the environment when the system on the drone is abnormal by actively creating vibration in the roll axis. Figure 8 shows the measured

FFT signal (x–y–z) of the IMU angle (Figure 8a) and angular acceleration (Figure 8b) to obtain the vibration signal of the proposed quadcopter flight operation. From the measured vibration harmonics in Figure 8, the harmonic frequencies of the vibration are determined as 1–20 Hz.

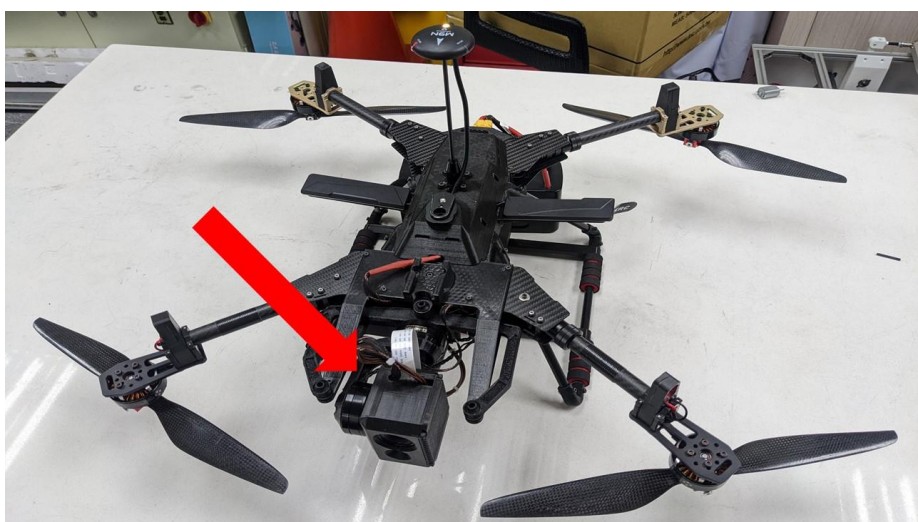

**Figure 6.** Proposed quadcopter.

**Table 1.** Specification of the quadcopter.

| | |
|---|---|
| Model weight | 4100 g |
| Number of rotors | 4 |
| Frame size | 690 mm |
| Power | 22.8~26.0 V |
| Average current | 15 A |
| Propeller | Diameter 14 inch 5.5 pitch |
| Max thrust | 2100 g |
| Average thrust | 1030 g |
| Hover thrust | 1025 g |
| Idle thrust | 50 g |

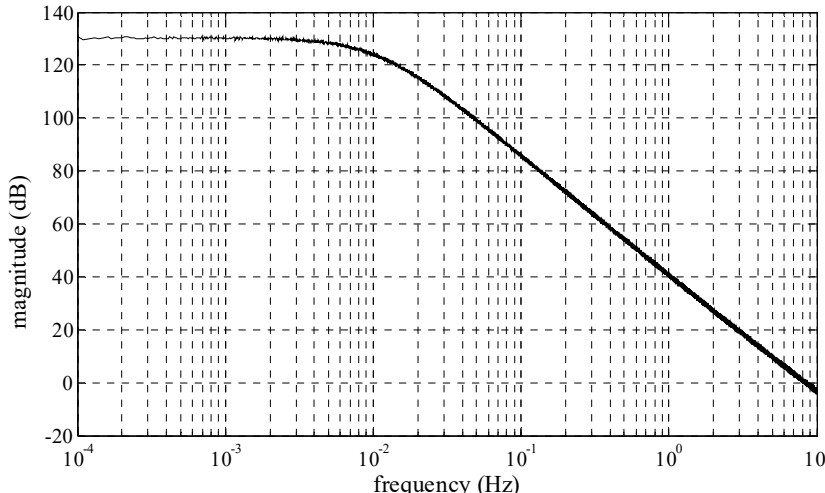

**Figure 7.** Measured frequency response of the controlled plant of the illustrated example.

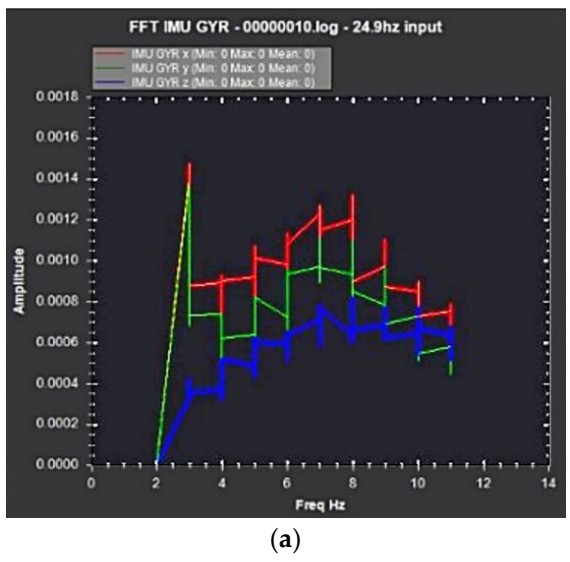

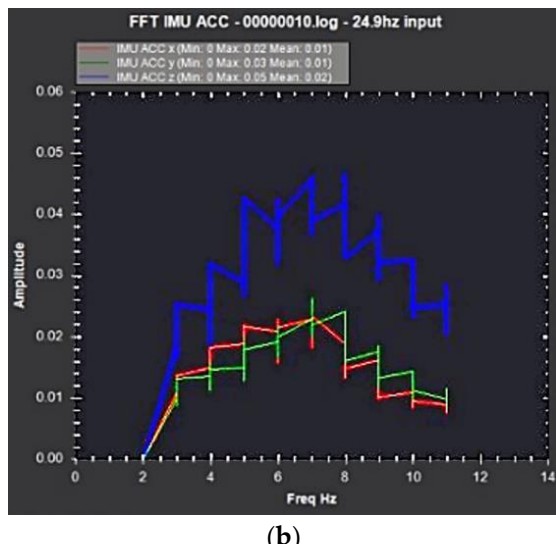

(**a**)　　　　　　　　　　　　　　　　　　　(**b**)

**Figure 8.** Measured vibration signal of the proposed quadcopter flight operation. (**a**) The measured FFT signal of the IMU angle and (**b**) the measured FFT signal of the IMU angular acceleration.

From the above analysis results, a disturbance with a period $T_d = 2$ s is given, and the dynamic stiffness of the proposed quadcopter control system is shown in Figure 8. From Figure 9a, it can be found that the magnitudes of the direct dynamic stiffness with the repetitive controller are larger than those without the repetitive controller, and the maximum of the direct dynamic stiffness can be found at harmonics of the periodic signal, i.e., $\frac{k}{T_d} = \frac{k}{2} = 0.5k$, $k = 1, 2, 3, \cdots$. From Figure 9b, the quadrature dynamic stiffness with $Q(s)$ is relatively smooth compared to that without $Q(s)$, which can be used to verify the stability being improved with $Q(s)$. Figure 10 shows the comparisons of loss factor, where the magnitudes of loss factor of the quadcopter control system with the repetitive controller are lower than those without the repetitive controller.

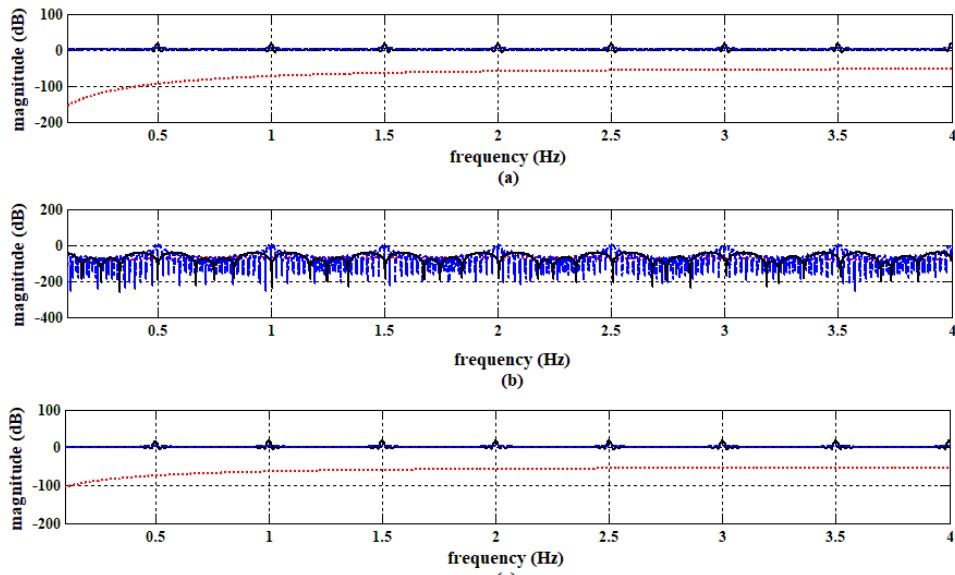

**Figure 9.** Magnitudes of dynamic stiffness for $T_d = 2$ s (red dotted line for Figure 2, blue dashed line for Figure 3, and black solid line for Figure 4): (**a**) magnitude plot of the direct dynamic stiffness, (**b**) magnitude plot of the quadrature dynamic stiffness, and (**c**) magnitude plot of the dynamic stiffness.

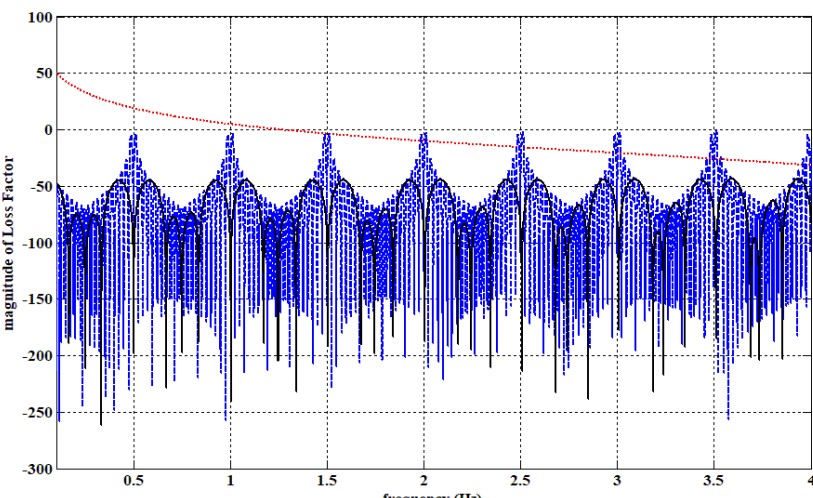

**Figure 10.** Magnitudes of loss factor for $T_d = 2$ s (red dotted line for Figure 2, blue dashed line for Figure 3, and black solid line for Figure 4).

Figures 11 and 12 shows the comparisons of the dynamic stiffness and loss factor with a periodic disturbance of a period $T_d = 0.5$ s. Similar to the results of Figures 9 and 10, it can be found that the dynamic stiffness with the repetitive controller is larger than that without the repetitive controller. The maximum of the direct dynamic stiffness can be found at harmonics of the periodic signal, i.e., $\frac{k}{T_d} = \frac{k}{0.5} = 2k, k = 1, 2, 3, \ldots$. In Figure 12, the loss factor of the quadcopter control system with the repetitive controller is lower than that without the repetitive controller.

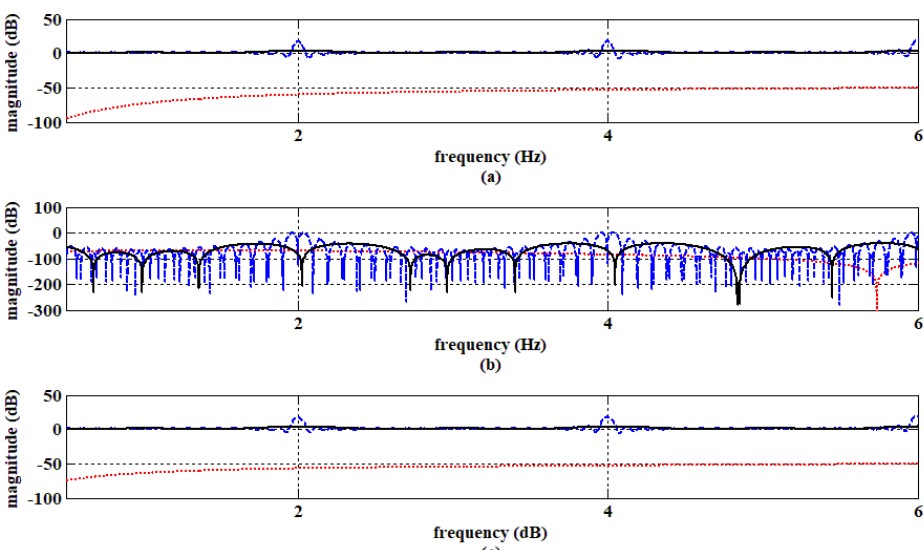

**Figure 11.** Magnitudes of dynamic stiffness for $T_d = 0.5$ s (red dotted line for Figure 2, blue dashed line for Figure 3, and black solid line for Figure 4): (**a**) magnitude plot of the direct dynamic stiffness, (**b**) magnitude plot of the quadrature dynamic stiffness, and (**c**) magnitude plot of the dynamic stiffness.

Based on Figure 4 with the given parameters, a periodic disturbance is given as

$$\begin{bmatrix} d(t) = \sin(2\pi t) + \sin(4\pi t)[u(t) - u(t-5)] + \\ \sin(8\pi t)[u(t-3) - u(t-6)] + \sin(10\pi t)[u(t-6) - u(t-20)] \end{bmatrix} \tag{30}$$

Based on the disturbance input in (30), as shown Figure 13a, the error responses of the control systems in Figures 2 and 4 can be obtained in Figure 13b. It can be found that the

error of Figure 4 is smaller than that of Figure 2, and a repetitive error marked blue dashed line can be found in the control system of Figure 2. The rapid decay rate of Figure 4 can be found, where the larger error can be found at the commands' switched time point, i.e., 0 s, 3 s, 5 s, and 6 s.

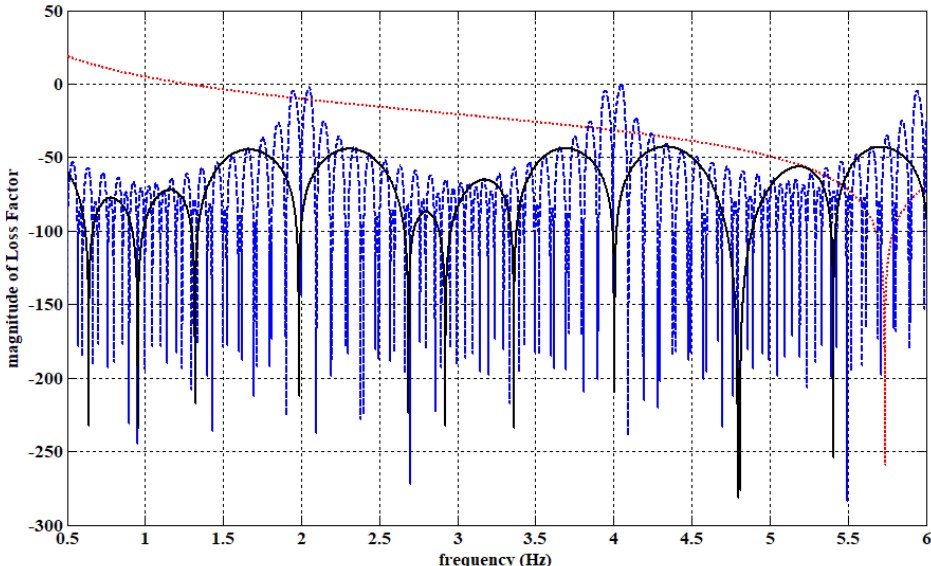

**Figure 12.** Magnitudes of loss factor for $T_d = 0.5$ s (red dotted line for Figure 2, blue dashed line for Figure 3, and black solid line for Figure 4).

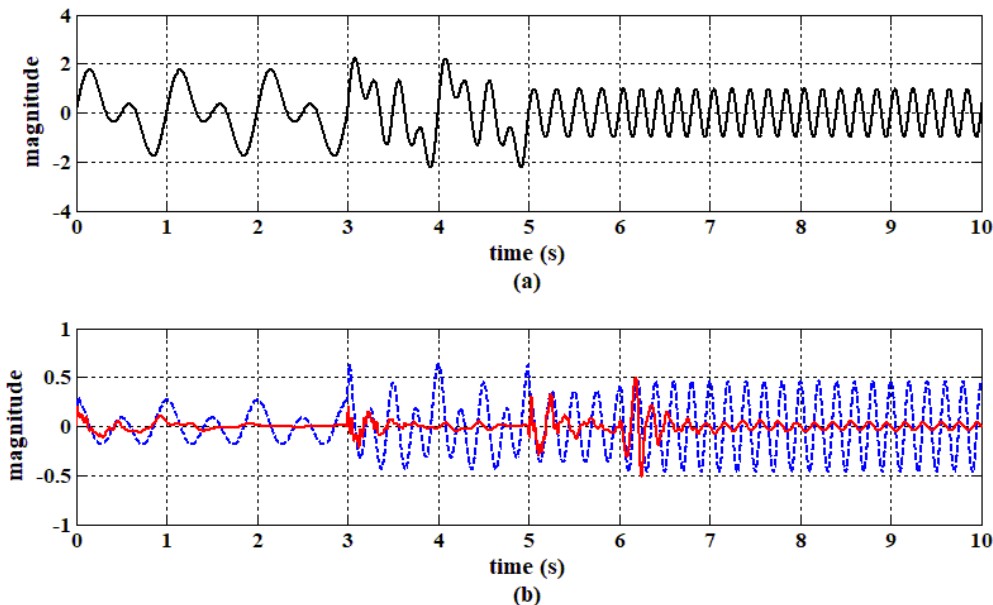

**Figure 13.** (**a**) Magnitude plot of the given disturbance of (30) and (**b**) the error responses of Figure 2 (blue dashed line) and Figure 4 (red solid line).

For the dynamic stiffness with the general periodic disturbance, considering the measured vibration harmonics in Figure 8, the harmonic frequencies are determined as 1–20 Hz, i.e., $T_d = 0.05$ s $\sim 1$ s. From Figure 14a, it can be found that the magnitudes of the direct dynamic stiffness with the repetitive controller are larger than those without the repetitive controller. From Figure 14b, the quadrature dynamic stiffness with $Q(s)$ is relatively smooth to compare that without $Q(s)$, which can be used to verify the stability being improved with $Q(s)$. Figure 15 shows the comparisons of loss factor, where the magnitudes of the loss factor of the quadcopter control system with the repetitive controller

are lower than those without the repetitive controller. Based on Figure 5 with the given parameters, a multi-periodic disturbance is given in Figure 16a, i.e.,

$$d(t) = \sin(4\pi t) + 2\sin(10\pi t) + 0.5\sin(14\pi t) \tag{31}$$

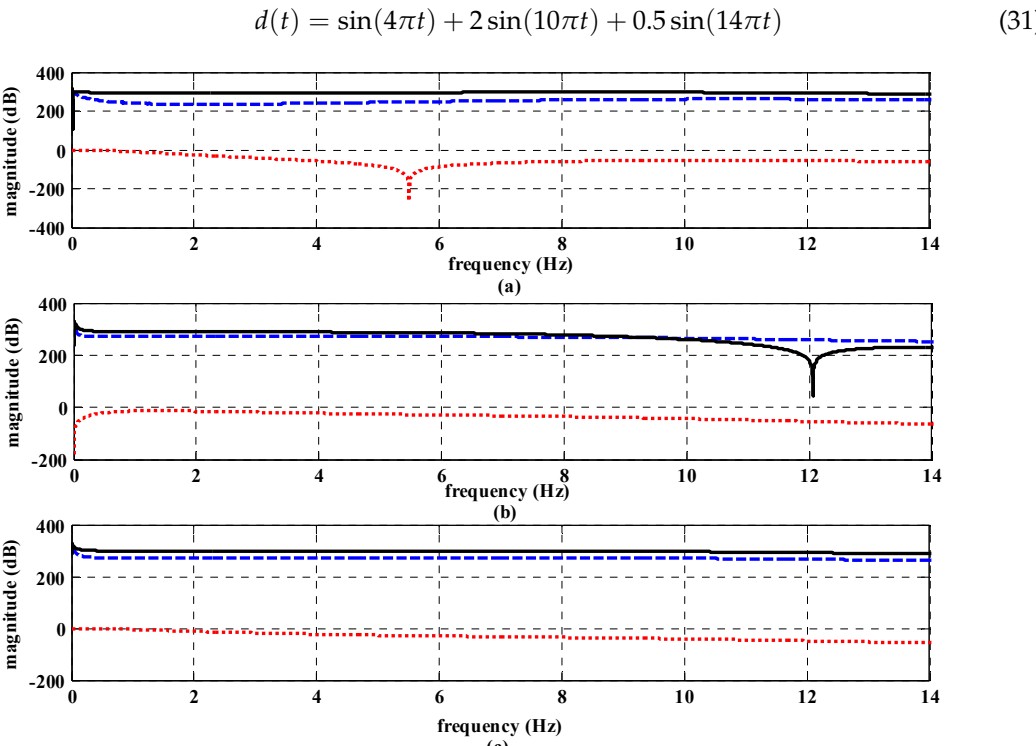

**Figure 14.** Magnitudes of dynamic stiffness for $T_d = 0.05$ s $\sim 1$ s (red dotted line for Figure 2, blue dashed line for Figure 5 without $Q(s)$, and black solid line for Figure 5): (**a**) Magnitude plot of the direct dynamic stiffness, (**b**) magnitude plot of the quadrature dynamic stiffness, and (**c**) magnitude plot of the dynamic stiffness.

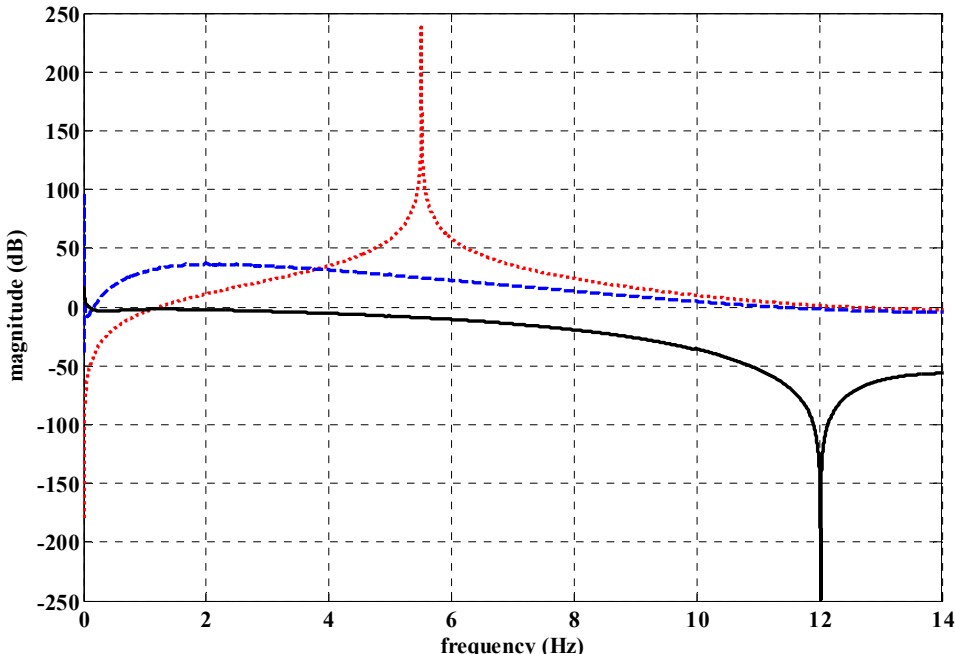

**Figure 15.** Magnitudes of loss factor for $T_d = 0.05$ s $\sim 1$ s (red dotted line for Figure 2, blue dashed line for Figure 5 without $Q(s)$, and black solid line for Figure 5).

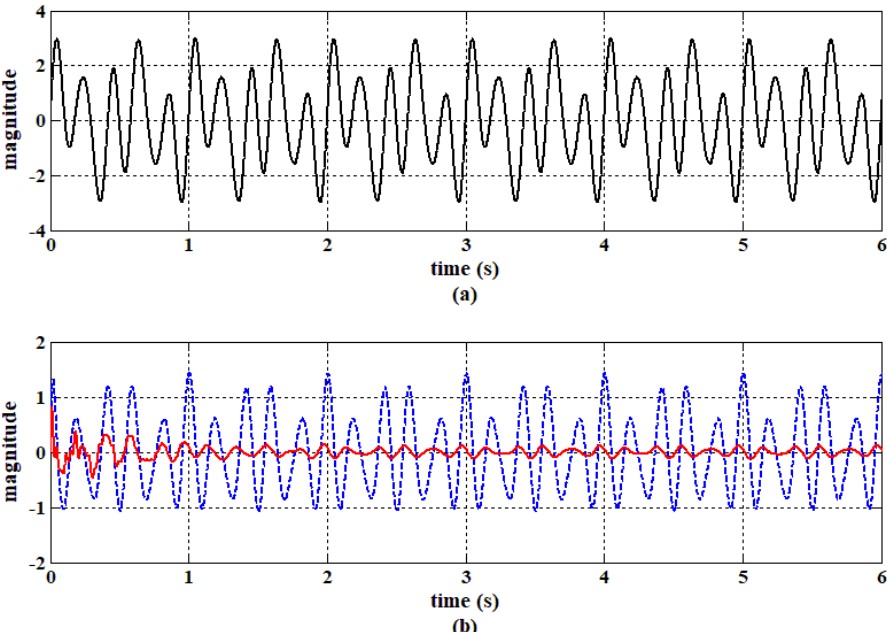

**Figure 16.** (**a**) Magnitude plot of the given disturbance of (31) and (**b**) the error responses of Figure 2 (blue dashed line) and Figure 5 (red solid line).

Based on the disturbance input in (31), the error responses of the control systems in Figures 2 and 5 can be obtained in Figure 16b. It can be found that the error of Figure 5 is smaller than that of Figure 2, and a repetitive error marked by the blue dashed line can be found in the control system of Figure 2. The rapid decay rate of Figure 5 can be found in Figure 16b.

## 5. Experimental Results Analysis

Figure 17 is the experimental setup of the study with the illustrated example of Figure 6. To provide a vibration situation under the quadcopter flight operation, a camera stabilizer constructed in the quadcopter is used to provide the vibration source. The control algorithm is implemented by the 32-bit ARM with 168 MHz installed on the quadcopter, and the inertial measurement unit is used to feedback the angular velocity and acceleration of X–Y–Z. Three kinds of circumstances are given to verify the proposed method. Note that the experiment environment wind speed is 0 to 1 knot in calm weather.

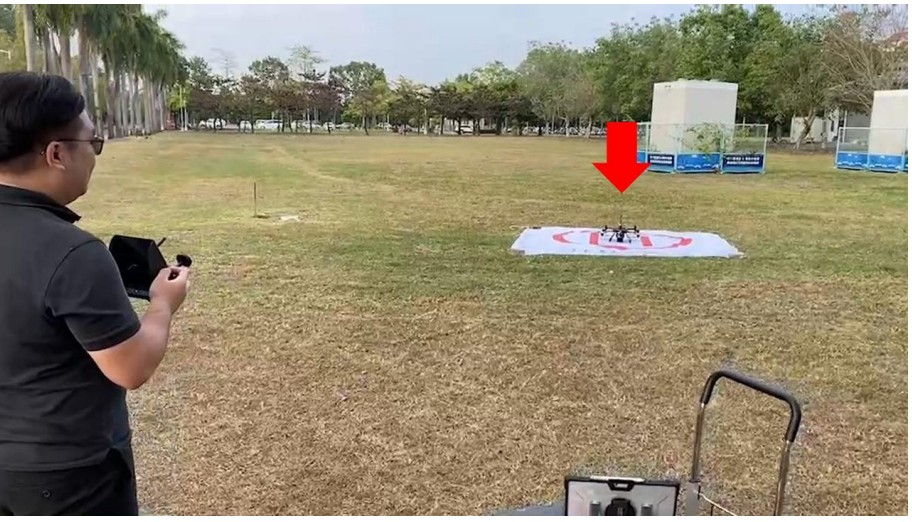

**Figure 17.** Experimental setup.

For the first case, the quadcopter control system only with feedforward + PID controller is given. The camera stabilizer is still, and any vibration is not provided by the stabilizer. Figures 18–21 show the experimental results under the stabilizer still. It can be found that the acceleration oscillations of the $x$-axis and $y$-axis are smaller than $\pm1$ m/s$^2$. The angle of oscillation is less than $\pm4$ degrees. A larger acceleration can be found at the time of taking off or landing of the quadcopter.

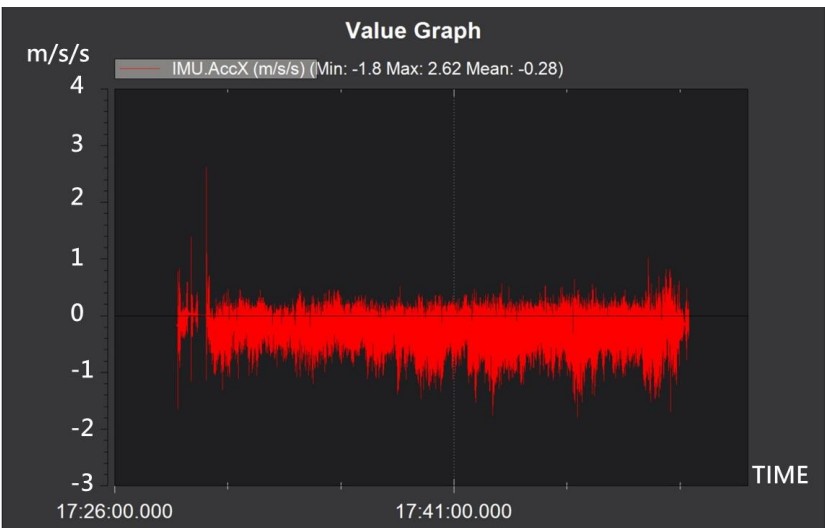

**Figure 18.** Acceleration response of $x$ axis under the stabilizer still.

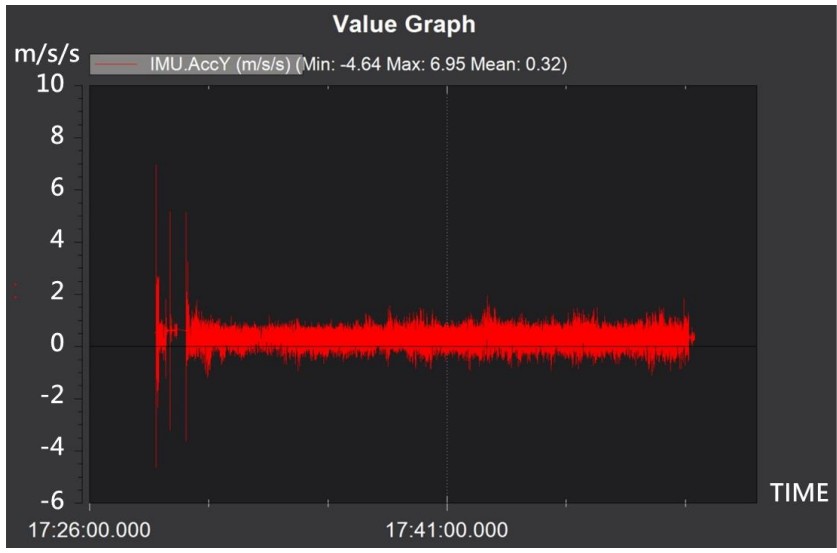

**Figure 19.** Acceleration response of $y$ axis under the stabilizer still.

For the second case, the vibration is produced by the stabilizer, and the repetitive controller is given in the quadcopter control system. Figures 22–25 show the experimental results. The acceleration oscillations of the $x$-axis and $y$-axis are smaller than $\pm1$ m/s$^2$. The angle of oscillation is less than $\pm5$ degrees.

For the third case, only with feedforward control + PID controller, the vibration is provided. Figures 26–29 show the experimental results. The acceleration oscillations of the $x$-axis and $y$-axis are larger than $\pm3$ m/s$^2$ and the angle of oscillation is less than $\pm10$ degrees. At 14:51:00:000, however, the quadcopter system almost becomes out of control, even though the controller seems to stabilize the quadcopter system. At 14:55:00:000, the accelerations are larger than $\pm40$ m/s$^2$ and the angles are larger than $\pm80$ degrees. This eventually leads to the crash of the experimental machine.

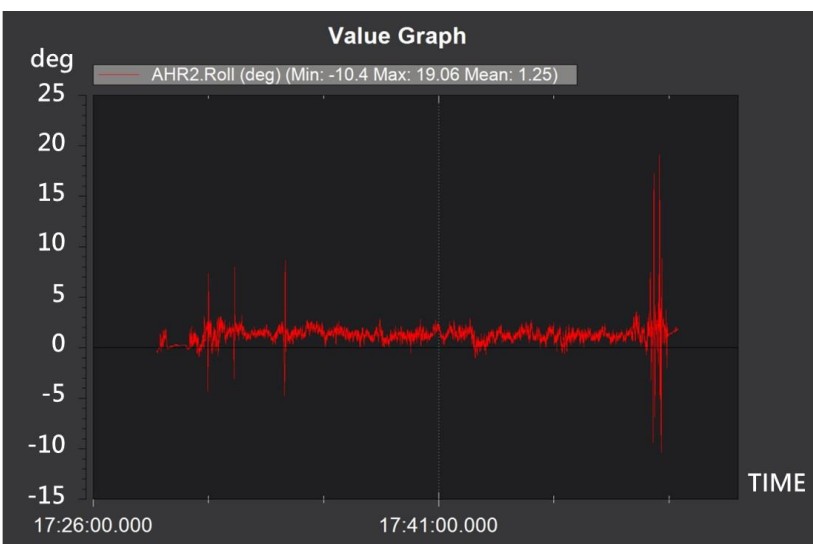

**Figure 20.** Roll angle response under the stabilizer still.

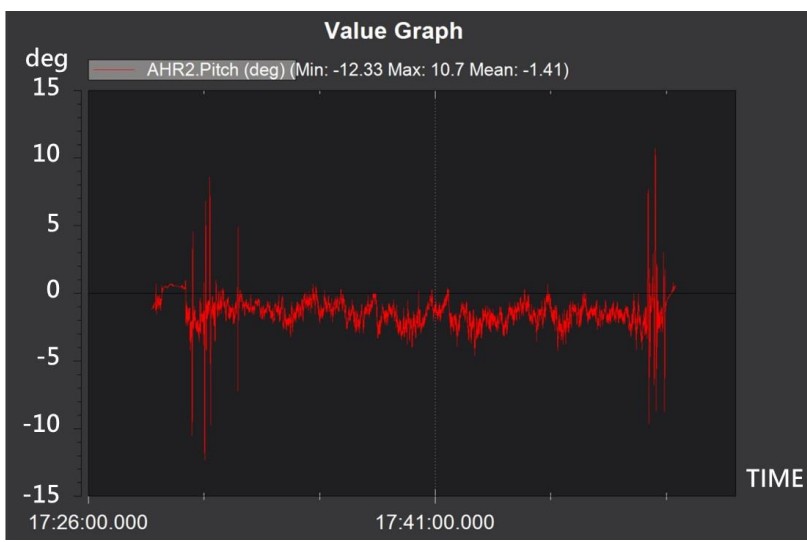

**Figure 21.** Pitch angle response under the stabilizer still.

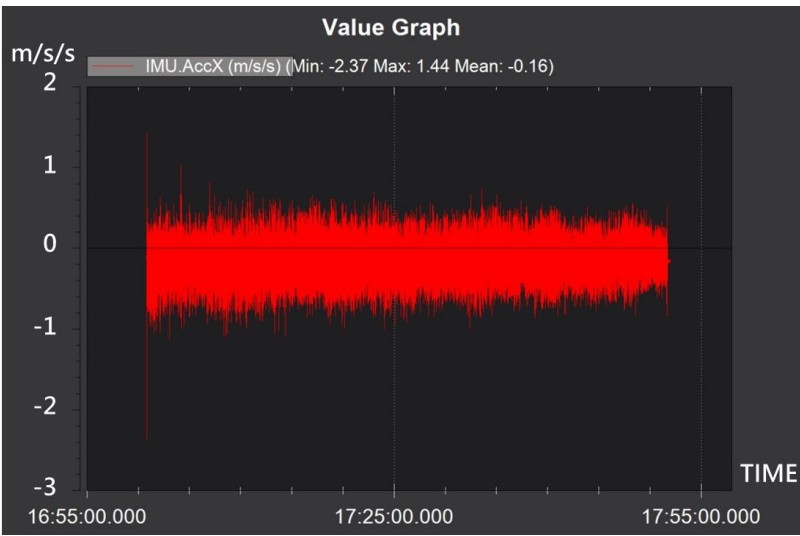

**Figure 22.** Acceleration response of X-axis with the repetitive controller under the generated vibration.

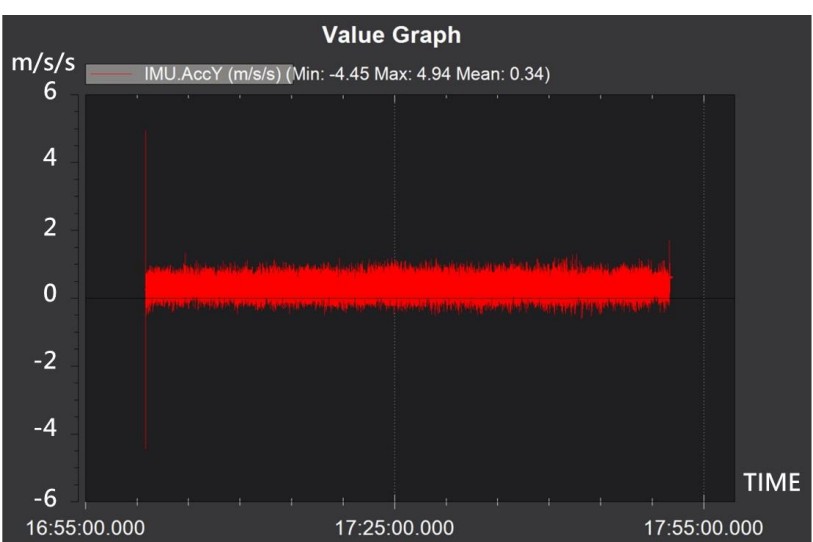

**Figure 23.** Acceleration response of Y-axis with the repetitive controller under the generated vibration.

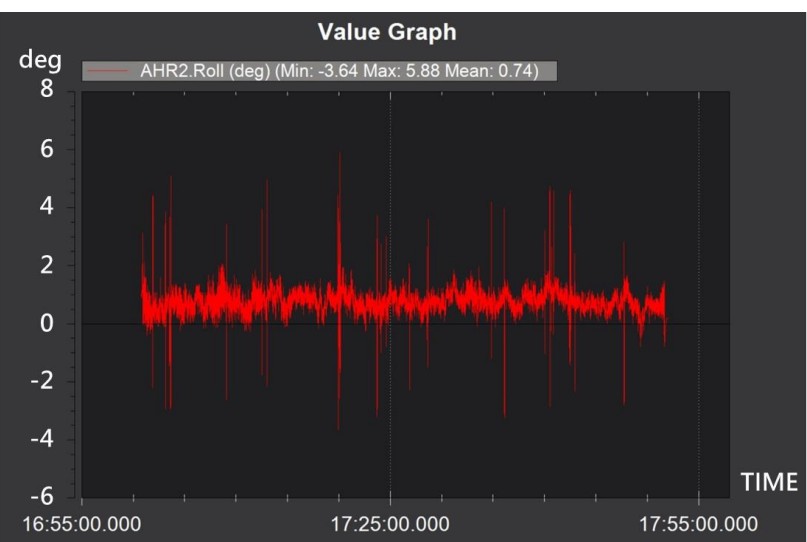

**Figure 24.** Roll angle response with the repetitive controller under the generated vibration.

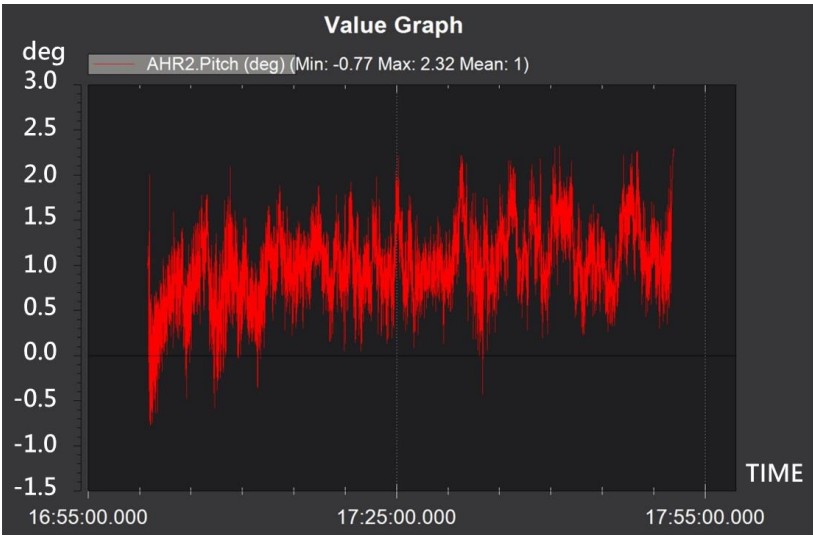

**Figure 25.** Pitch angle response with the repetitive controller under the generated vibration.

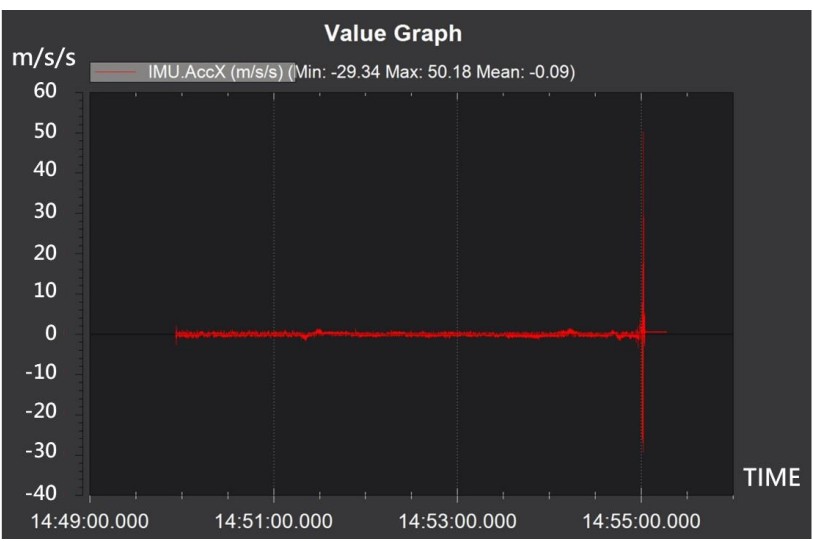

**Figure 26.** Acceleration response of *x*-axis without the repetitive controller under the generated vibration.

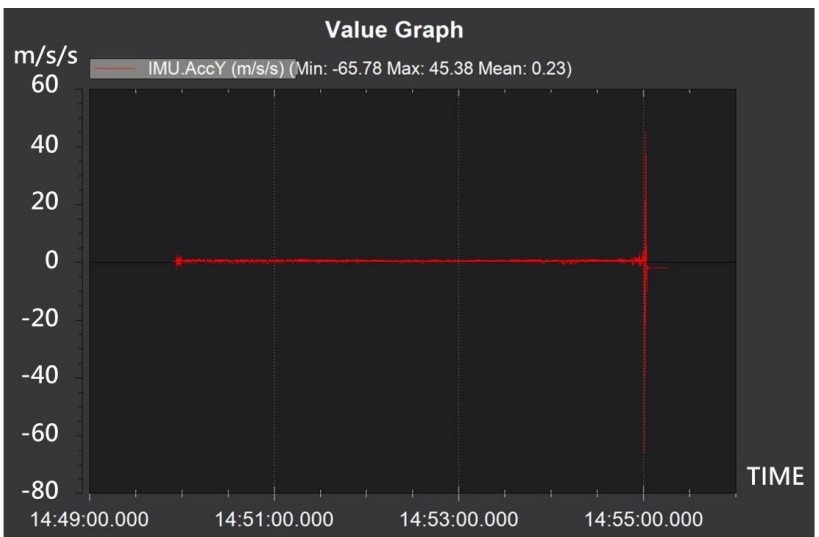

**Figure 27.** Acceleration response of *y*-axis without the repetitive controller under the generated vibration.

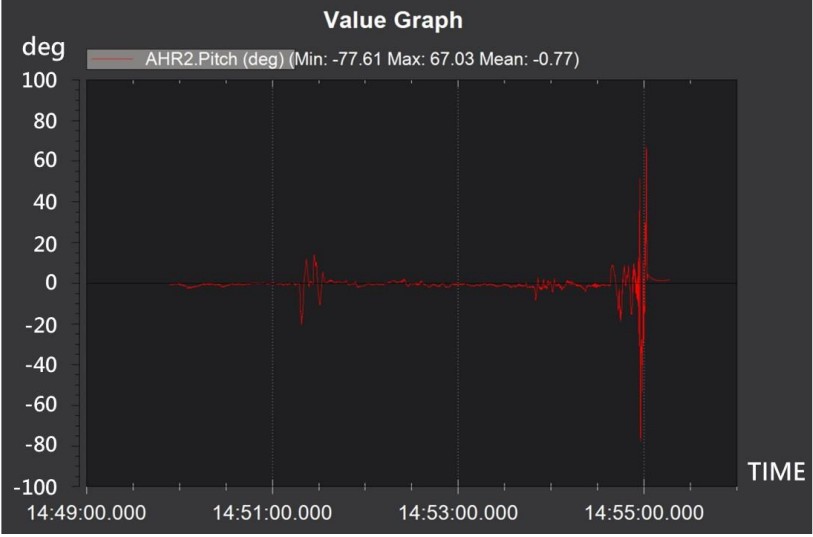

**Figure 28.** Pitch angle response without the repetitive controller under the generated vibration.

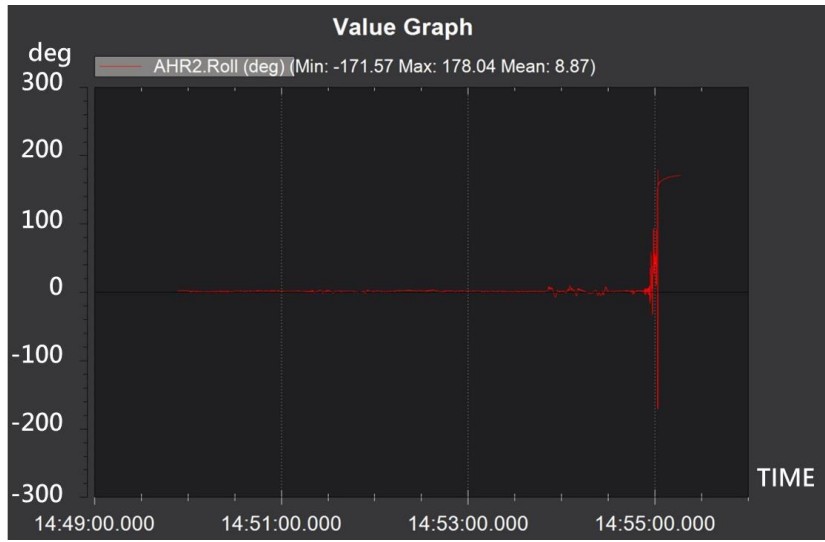

**Figure 29.** Roll angle response without the repetitive controller under the generated vibration.

## 6. Conclusions

This paper proposed the dynamic stiffness analysis method of the quadcopter control system with the repetitive controller. A theoretical model of the dynamic stiffness of the quadcopter control system is established through analyzing the definition of the dynamic stiffness. The direct dynamic stiffness and the quadrature dynamic stiffness with variable frequencies are used to verify the effect of the quadcopter control system. Simulated and experimental results show that the magnitudes of the direct dynamic stiffness with the proposed repetitive controller are larger than those without the repetitive controller. The quadrature dynamic stiffness with the proposed repetitive controller is relatively smooth compared to that without the proposed repetitive controller, which can be used to verify the stability being improved by the designed repetitive controller. In addition, the magnitudes of the loss factor of the quadcopter control system with the repetitive controller are lower than those without the repetitive controller.

**Author Contributions:** Conceptualization, W.-S.Y. and C.-Y.L.; methodology, W.-S.Y.; software, C.-Y.L.; validation, C.-Y.L.; formal analysis, W.-S.Y.; investigation, W.-S.Y.; resources, C.-Y.L.; data curation, C.-Y.L.; writing—original draft preparation, C.-Y.L.; writing—review and editing, W.-S.Y.; visualization, C.-Y.L. and W.-S.Y.; supervision, W.-S.Y. All authors have read and agreed to the published version of the manuscript.

**Funding:** This research received no external funding.

**Conflicts of Interest:** The authors declare no conflict of interest.

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
