# Peer review of "Dynamic Stiffness Enhancement of the Quadcopter Control System"

_electronics, doi:10.3390/electronics11142206_

Round 1

Reviewer 1 Report

In the manuscript “Dynamic Stiffness Enhancement of the Quadcopter Control System”, the authors proposed a dynamic stiffness analysis method of the quadcopter control system with the repetitive controller. They established a theoretical model of the dynamic stiffness of the quadcopter control system. The direct dynamic stiffness and the quadrature dynamic stiffness with variable frequencies are used to verify the effect of the quadcopter control system.

Given the expediency of providing a report, my comments are somewhat limited, though I hope they are still useful to the editors and authors:

The topic chosen by the authors is of importance and worthy of investigation. Overall the article is good, the methodology and procedure appear sound and the results are interesting. This paper can be accepted for publication after necessary revisions. The following issues must be addressed and clarified before acceptance of the article.

1.      The novelty of the work should be mentioned clearly in the abstract.

2.      Use either Fig. or figure to refer to the figures in the text

3.      Figure 8 both parts are of low resolution and not readable, replace them with high resolution figures. Similarly for figures 17-28.

4.    I noticed several grammatical/ sentence structuring errors throughout the paper.  A very few examples are: In line 11 ‘To rapid reduce’ must be ‘To rapidly reduce’, line 13 ‘frequencies variable’ must be ‘variable frequencies’, line  249 ‘to analysis’ must be ‘to analyze’, line 405 ‘frequencies variable’ must be ‘variable frequencies’, line 406 ‘results are shown that’ must be ‘results show that’ etc…..The English write up needs much improvement throughout the manuscript.

Reviewer 2 Report

The authors have proposed a dynamic stiffness analysis method of the quadcopter control system with a repetitive controller. They have presented simulation and experimental results to show the magnitude of the dynamic stiffness. The manuscript is organized well however there are a few minor comments:

1. Please provide the simulation details and also, comment on the assumption that have been considered for the  simulation.

2. How the proposed dynamic stiffness method is implemented to obtain the experimental results? Describe the implementation details.

3. Provide the comparison between simulation and experimental results, Also, if the experimental results can be compared to the traditional methods, please provide the comparison to demonstrate the better performance of the dynamic stiffness analysis method. 
